# SP1 and NFY Regulate the Expression of *PNPT1*, a Gene Encoding a Mitochondrial Protein Involved in Cancer [note 1]

**DOI:** 10.3390/ijms231911399

**Published:** 2022-09-27

**Authors:** Ignacio Ventura, Fernando Revert, Francisco Revert-Ros, Lucía Gómez-Tatay, Jesús A. Prieto-Ruiz, José Miguel Hernández-Andreu

**Affiliations:** Grupo de Investigación en Medicina Molecular y Mitocondrial, Universidad Católica de Valencia ‘San Vicente Mártir’, 46001 Valencia, Spain

**Keywords:** SP1, PNPT1, NFYA, mitochondria, liver cancer

## Abstract

The Polyribonucleotide nucleotidyltransferase 1 gene (*PNPT1*) encodes polynucleotide phosphorylase (PNPase), a 3′-5′ exoribonuclease involved in mitochondrial RNA degradation and surveillance and RNA import into the mitochondrion. Here, we have characterized the *PNPT1* promoter by in silico analysis, luciferase reporter assays, electrophoretic mobility shift assays (EMSA), chromatin immunoprecipitation (ChIP), siRNA-based mRNA silencing and RT-qPCR. We show that the Specificity protein 1 (SP1) transcription factor and Nuclear transcription factor Y (NFY) bind the *PNPT1* promoter, and have a relevant role regulating the promoter activity, *PNPT1* expression, and mitochondrial activity. We also found in Kaplan–Meier survival curves that a high expression of either PNPase, SP1 or NFY subunit A (NFYA) is associated with a poor prognosis in liver cancer. In summary, our results show the relevance of SP1 and NFY in *PNPT1* expression, and point to SP1/NFY and PNPase as possible targets in anti-cancer therapy.

## 1. Introduction

Mitochondria play a major role in the biology of eukaryotic cells as ATP producers through oxidative phosphorylation. Nevertheless, these organelles also work as a hub for metabolic functions, such as nutrient catabolism for energy production, the generation of biosynthetic precursors, redox homeostasis, and the management of metabolic waste (reviewed by [1]), and are involved in other cellular processes in health and disease (reviewed by [2]). Since mitochondria are organelles with a dual biogenesis, the transport of nuclear-encoded proteins and cytosolic RNA are essential processes for mitochondrial function. Unlike the well-characterized mitochondrial protein import, (reviewed by [3]), RNA transport into the mitochondria is a still a poorly understood process (reviewed by [4]).

The human Polyribonucleotide Nucleotidyltransferase 1 (*PNPT1*) gene, whose cytogenetic localization is 2p16.1, encodes Polynucleotide Phosphorylase (PNPase) which is essential for cell survival and mitochondrial DNA maintenance [5,6]. PNPase participates in a wide array of cellular processes, including nucleus-encoded RNA import into mitochondria, the processing and decay of miRNA and mRNA [7], and the processing of polycistronic mitochondrial transcripts and tRNAs [8]. Although PNPase was discovered in the 1950s [9], new functions of this protein have been recently discovered [10,11]. Once synthesized, PNPase is guided to the mitochondrial intermembrane space (IMS) [6,12] by a mitochondrial targeting sequence which is cleaved off upon translocation into the IMS [13]. Within the IMS, PNPase binds and facilitates the mitochondrial import of RNase P protein, 5S rRNA, and RNase MRP RNAs [5]. In *E. coli*, PNPase also plays a role in RNA turnover and in the degradation of oxidized RNA [14]. Interestingly, upon apoptosis initiation, PNPase is released to the cytosol where it triggers the decay of mRNA and polyadenylated noncoding RNA [15]. In the cytoplasm, PNPase degrades c-MYC mRNA resulting in cell growth arrest [16]. A stable PNPase knockout in mouse embryonic fibroblasts resulted in the loss of both mitochondrial DNA and cellular respiration [7]. In addition, *PNPT1* mutations impairing mitochondrial import caused early embryonic lethality in zebrafish [5]. PNPase translocation into the mitochondrial IMS is regulated through binding to the oncoprotein TCL1 in the cytoplasm. TCL1 prevents the mitochondrial localization of PNPase and thus leads to the switch from OXPHOS to glycolytic metabolism, which is a feature of stem and cancer cells [17,18,19]. Furthermore, the LIRPPRC/SLIRP complex suppressed 3′ exonucleolytic mRNA degradation mediated by PNPase and SUV3 [20], the main RNA-degrading complex in human mitochondria [21], showing the important role of PNPase in the degradation and surveillance of mitochondrial RNA.

According to clinical reports, *PNPT1* mutations are present in patients suffering from oxidative phosphorylation deficiency [22,23,24], autoinflammatory syndrome [25], autosomal recessive deafness (DFNB70) [26], delayed myelinization [27] and Leigh syndrome [8]. Furthermore, patients with reduced PNPase activity exhibit dsRNA accumulation and elevated interferon levels in serum [24]. Besides, *PNPT1* inactivation in breast cancer cells leads to increased c-Myc mRNA levels and consequent radio resistance [28].

*PNPT1* gene maps to 2p15–2p16.1 in which alterations such as deletion and amplification are involved in human cancers, such as diffuse large B-cell lymphoma [29] and various genetic disorders [30]. Interestingly, a *PNPT1*–*ALK* fusion gene generated by anaplastic lymphoma kinase gene (*ALK*) rearrangement (fusion of exons 1–19 of *PNPT1* with exons 19–29 of *ALK*) exerted significant benefit to crizotinib treatment in a NSCLC (non-small cell lung cancer) patient [31]. PNPase was identified as a tumor-associated antigen expressed by acute lymphoblastic leukemia-derived dendritic cells, suggesting this protein is a possible target for immunotherapy in this tumor type [32].

Understanding the regulation of the promoter activity of a critical gene is necessary to determine the relevance of the different transacting factors that could play a critical role in pathogenesis and prognosis of a related disease. The *PNPT1* promoter was initially characterized as TATA-less and CAAT box-lacking [33], and embedded in a GC-rich sequence. Putative binding sites for ISRE, GAS, IRF-1 and SP1 were identified, and the promoter was shown to be responsive to IFN Type 1 through the ISRE element, suggesting a possible role for *PNPT1* in growth regulation, terminal differentiation and/or cellular senescence. A similar approach identified additional putative binding sites mainly for housekeeping (ATF, CAAT, NFY, TAXCREB, NF1) and interferon responsive (HMGIY, IRF7) transcriptional elements [34].

Nevertheless, except for ISGF3 (IFN-Stimulated Gene Factor 3) and STAT2/IRF9 (Signal Transducer and Activator of Transcription 2-IFN Regulatory Factor 9) [11], no other transcription factor has been characterized so far for the *PNPT1* promoter, despite the importance of this gene for mitochondrial and cellular function. In this study, we have carried out an exhaustive functional characterization of the *PNPT1* promoter and determined the role of two important transcription factors, SP1 and NFYA, for *PNPT1* expression. Our results could shed light on the functions of this protein in health and disease.

## 2. Results

### 2.1. Predicted Cis-Elements of PNPT1 Promoter

A 404-bp DNA sequence encompassing the human *PNPT1* promoter (nucleotides 4692 to 5095 of NG_033012.1 NCBI Reference sequence) was analyzed with MatInspector and Promo online applications with default settings. The analysis revealed an Interferon-Sensitive Response Element (ISRE), a putative binding site for NFY transcription factor (CCAAT) [35], and a GC-box (CCGCCC) overlapping a neuron-restrictive silencer element (NRSE)-like sequence (consensus 5′-TTCAGCACCACGGACAGCGCC-3′) [36]. The GC-box was part of a putative SP1 transcription factor binding site [consensus 5′-(G/A)(C/T)(C/T)CCGCCC (C/A)-3′] [37] (Figure 1A). Transcription starting sites are also shown (TSS) [38].

The ISRE, SP1 and NFY binding sites were almost fully conserved in rabbit, cow and sheep genomes, which also displayed NRSE-like sequences similar to the reported consensus (5′-TTCAGCACCACGGACAGCGCC-3′) [36] (Figure 1B).

### 2.2. Analysis of Cis-Elements with a PNPT1-Promoter Construct

The contribution of individual DNA elements to *PNPT1* promoter activity was assessed with HeLa cells using a firefly luciferase reporter construct containing a 396-bp *PNPT1* promoter region (Figure 2), and with derived constructs holding point mutations of the four predicted elements. Cells were transiently transfected and the luciferase activity in lysates was measured. Maximum promoter activity was assessed by dose-response analysis; cell viability after transfection was also evaluated (not shown).

The activity of the *PNPT1* promoter was similarly affected by single mutations in ISRE, CCAAT (NFY) or SP1 motifs (Figure 2), which downregulated promoter activity by ~60%. Meanwhile, a point mutation in NRSE-like sites reduced the promoter activity by ~25%. A double mutant construct containing mutations in both SP1 and NFY motifs showed an activity similar to that of the singly mutated counterparts, suggesting that these elements work in a coordinated fashion. The mutation of one of these elements could be affecting the activity of the other, and for this reason the activity of the double SP1+NFY motif mutants is similar to that of single (SP1 or NFY motif) mutants.

Interestingly, the mutating ISRE motif further reduced the activity of NFY mutants to residual levels, suggesting the promoter could be working with two transcriptional complexes, one for ISRE and a second one for SP1/NFY motifs. The overlapping of SP1 and NRSE elements raises the possibility that the mutation in the latter could be affecting the SP1 assembly rather than the binding of a specific trans-element to this motif.

### 2.3. SP1 and NFYA Bind PNPT1 Promoter

To determine whether the predicted cis-elements of the *PNPT1* promoter were bound by their putative transcription factors, EMSA assays with labeled double-stranded oligonucleotide probes representing the identified motifs were performed.

Since SP1 and NRSE cis-elements overlap within the *PNPT1* promoter, one of the used probes encompassed both motifs (Figure 1). The three probes mixed with HeLa nuclear extracts (basal conditions) bound to high molecular weight complexes (Figure 3A), suggesting that the nuclear extracts contained trans-elements recognizing the proposed motifs.

The previously described and characterized ISRE motif [11], was not included in our assays.

To further characterize the specificity of complex binding to the motifs, labeled probes were competed with unlabeled, double-stranded wild-type and mutated oligonucleotides. Mutation of SP1 and NFY motifs in the unlabeled oligonucleotides impaired their capacity to complete the formation of complexes of the labeled wild-type probe with nuclear factors (Figure 3B,C), indicating the existence of nuclear proteins that specifically bind to these motifs. Besides, a mutation in the NRSE-like (NRSE) motif did not affect the formation of probe-protein complexes, suggesting that NRSE-specific trans-elements are not present in the complex, or that the NRSE motif is not really functional.

The identity of trans-acting elements was confirmed with specific antibodies. Thus, the use of αSP1 but not αNRSE hampered probe-protein complex formation, suggesting that SP1, and not NRSE, binds the corresponding motif in HeLa cells (Figure 3B). Likewise, antibodies against NFY subunit A (αNFYA) produced higher molecular weight complexes (supershift), which were not observed using antibodies to IRF-2, and which can be considered as control antibodies in this case (Figure 3C). All of the above indicates that SP1 and NFYA polypeptides bind their predicted elements within the *PNPT1* promoter in vitro.

To investigate the in vivo relevance of SP1 and NFYA binding to the *PNPT1* promoter (Figure 3D), ChIP assays were performed using HeLa cells. Both αSP1 and αNFYA enriched the amount of the *PNPT1* promoter present in the precipitated material compared to controls, which is indicative of SP1 and NFYA binding to the *PNPT1* promoter within the genomic DNA of cultured HeLa cells. Furthermore, similar results that confirm these findings can be found in online databases (Appendix A).

Collectively, these results strongly suggest that SP1 and NFYA are part of the transcriptional machinery driving the *PNPT1* expression.

### 2.4. PNPT1 Promoter Regulation by SP1 and NFYA in HeLa Cells

To evaluate the impact of SP1 and NFYA transcription factors on *PNPT1* expression, we performed transfections with SP1- and NFYA-specific siRNAs followed by RT-qPCR analysis (Figure 4A). SP1 and NFYA silencers displayed significant and similar efficacies at 10 nM and 50 nM, respectively, in reducing the expression of their target genes. As expected, SP1 silencing runs in parallel with a reduction of *PNPT1* expression. Interestingly, SP1 silencing also produced a slight rise of NFYA expression (and vice versa), which could not counteract the downregulation of SP1 on *PNPT1* mRNA levels. On the contrary, siRNA-mediated NFYA downregulation did not reduce *PNPT1* expression. This could be the consequence of the parallel rise of SP1 levels observed in NFYA-silenced HeLa cells (Figure 4A), an insufficient reduction of NFYA activity, the presence of other transacting factors that could bind the same motif (i.e., CCAAT enhancer binding proteins or C/EBPs), or a posttranslational compensatory stabilization of *PNPT1* mRNA not observed with the reduction in SP1 levels.

To analyze the relevance of SP1 and NFYA expression in the transcriptional activity of the *PNPT1* promoter (Figure 4B), we used SP1 (10 nM) and NFYA (50 nM) siRNAs in luciferase reporter assays with the wild-type 396 bp *PNPT1* promoter construct. Both SP1 and NFYA silencers reduced luciferase activity, although the effect of the SP1 silencer was 3-fold higher than that of the NFYA counterpart. These results indicated that both SP1 and NF-Y factors control, at least in part, the *PNPT1* promoter basal activity in HeLa cells.

### 2.5. PNPT1, SP1 and NFYA Overexpression Is Correlated with a Poor Prognosis in Liver Cancer Patients

There is growing evidence suggesting that mitochondrial activity is an essential factor in cancer progression, and not just as a source of pathogenic stress. In this sense, mitochondrial activity is associated with hepatocellular cancer prognosis. Interestingly, liver cancer patients with higher PNPase levels showed a poorer prognosis than patients with a lower expression. In addition, high SP1 and NFYA expression levels were evidenced as being linked to a worse outcome. On the contrary, the transacting factors which bind ISRE motif in the *PNPT1* promoter did not relate to disease progression (Figure 5).

These results suggested that the activity of the SP1/NFY transcriptional complex on the *PNPT1* promoter could be relevant for tumor growth and for the survival of liver cancer cells; they also suggested that there must be a pathological context in which the activity of the *PNPT1* promoter depends on both SP1 and NFY, rather than just SP1. Consequently, the regulation of the formation of the SP1/NFY complex could be the underlying cause for the rise in the *PNPT1* expression associated with a poor outcome in liver cancer patients.

## 3. Discussion

Promoters are critical regulatory elements in physiological and pathological conditions. Although they have not been considered as good pharmacological targets, several promising approaches to produce chemical probes and drugs targeting trans-acting factors have been proposed [39].

Our analysis of the *PNPT1* promoter identified the previously characterized ISRE motif and additional SP1 and NFY cis-acting sites. These elements are conserved among mammals, pointing out their relevance in the regulation of *PNPT1* expression. The functionality of these motifs has been demonstrated by mutagenesis using luciferase reporter plasmids. Besides, EMSA assays showed in vitro binding of NFYA and SP1 transcription factors to the corresponding putative sites on *PNPT1*, and in vivo binding was confirmed by ChIP assays.

Both SP1 [40] and NFY [41] are involved in the expression of mitochondrial proteins, and they have been reported to work in association [42], and even to physically interact [43]. In this sense, it has been reported that SP1 and NFY could operate as a unique trans-acting complex when their binding sites are in close proximity [44], as we found in the *PNPT1* promoter. Our results showed that the mutation of both NFY and SP1 binding elements did not reduce the promoter activity beyond that displayed by single mutants of NFY or SP1 elements. However, additional mutation on ISRE caused a further reduction in activity. Collectively, these results suggested that the *PNPT1* promoter operates with two transcriptional complexes, one for the ISRE motif, and a second one for the NFY/SP1 sites.

The ISRE motif binds a transcriptional complex which is active in basal conditions and responds to interferon γ by increasing transcriptional activity. Our results strongly suggest that the second complex operates under the control of SP1 in cancerous cells growing in conventional unstimulated conditions. In these conditions, NFY binds the promoter, but do not critically control its activity. In addition, we found a reduction of *PNPT1* levels secondary to SP1 silencing, reinforcing the idea that the transcription factor SP1 regulates *PNPT1* expression. However, in our experimental conditions, we did not observe a decrease in *PNPT1* mRNA levels when NFYA was silenced, in contrast to the recombinant promoter activity which was downregulated upon NFYA silencing. The apparent discrepancy between the effect of NFYA silencing on native and recombinant promoter activities could be a consequence of the relatively high number of promoter copies present in cells transfected with the reporter construct, which reduces the ratio of available transacting factors. Moreover, the inefficacy of NFYA silencing to reduce *PNPT1* mRNA levels could also be due to a compensatory stabilization of *PNPT1* mRNA rather than to an absence of NFYA activity on the promoter. This compensatory mechanism could be dependent on SP1 but not on NFY factors.

The relevance of the *PNPT1* promoter regulation in vivo can be deduced from the analysis of Kaplan–Meier survival plots. We found that a high PNPase expression in the tumor cells of hepatocellular cancer patients is linked to a poor prognosis. In addition, we found that SP1 or NFYA levels were also associated with a poor outcome, whereas the levels of trans-acting elements of the ISRE motif did not show a significant relation with liver cancer progression. In this sense, dysregulation of mitochondrial biogenesis has been linked to a hepatocellular carcinoma.

SP1 is overexpressed in a number of cancer types and is considered as a poor prognosis factor [45], and NFY has been reported to be overexpressed in lung [46] and hepatocellular cancer [47]. Our results suggest that in the context of tumor cell cultures, SP1 shows a more relevant role than NFY in driving *PNPT1* expression. However, in the context of tumors of hepatocellular cancer, both NFY and SP1 seem to be linked to PNPase overexpression and to the outcome of patients. This suggests that the tumor microenvironment of patients has a key influence on the gene expression pattern of tumor cells, which is not mimicked by cultured cells (Appendix A).

Further studies are necessary to determine whether, under liver cancer pathological conditions, the accumulation of PNPase is a consequence of SP1/NFY complex formation or just a matter of expression levels of the individual transcription factors. The characterization of transcriptional complexes could allow the identification of target-specific molecules and the development of novel pharmacological repertoires for the therapeutic regulation of the transcription of genes such as *PNPT1*, whose overexpression is associated with serious diseases like liver cancer.

## 4. Materials and Methods

### 4.1. Analysis of the Promoter Region of the PNPT1 Gene

The proximal promoter sequence of the *PNPT1* gene was obtained from the NCBI website. Nucleotides 4692 to 5095 of NG_033012.1 NCBI reference sequence were analyzed with MatInspector (Genomatix, Munich, Germany) and Promo (Alggen, Barcelona, Spain) online applications.

### 4.2. Cell Culture, Mitochondrial Activity and Quantification

HeLa cells were grown in Eagle’s Minimum Essential Medium (EMEM; GibcoBR) supplemented with 10% fetal bovine serum (Gibco, Waltham, MA, USA) 2 mM glutamine (Gibco, Waltham, MA, USA) and 1% penicillin-streptomycin (PS; 10,000 U/mL; Gibco). Cells were maintained at 37 °C in a humidified atmosphere with 5% CO_2_.

Ninety-six-well plates were used to assess mitochondrial activity and determine the number of mitochondria per well in gene-silenced cells.

Mitochondrial activity was determined by adding resazurin solution (10% *vol/vol*) (Biotium, Fremont, CA, USA) to culture medium, and relative quantification of mitochondria was performed with MitoTracker Green FM at 150 nM in PBS (ThermoFisher Scientific, Waltham, MA, USA), following manufacturer’s indications. Resazurin reduction and MitoTracker green fluorescence emission were quantified using a Victor X5 fluorimeter (Perkin Elmer, Melbourne, Australia).

Both procedures display a direct linearity between 1000 and 10,000 cells per well.

### 4.3. DNA Constructs

The 396 bp sequence upstream of the ATG of *PNPT1* and derived mutants were synthesized and cloned into the KpnI and HindIII sites of the pUC57 plasmid (Genescript, Piscataway, NJ, USA).

Wild-type and mutated promoters were subcloned into KpnI/HindIII of pGL4-Luc luciferase reporter vector (Promega, Sydney, Australia). DH5-competent bacteria were transformed with pGL4-Luc constructs and grown in selective LB Agar ampicillin (100 g/mL) plates. Isolated colonies were grown in selective LB medium supplemented with amplicilin. PureYield™ Plasmid Miniprep and Midiprep kits (Promega) were used for the purification of plasmids. The fidelity of the promoter region of all clones was verified by sequencing. Sequences of mutants are shown in Table 1.

### 4.4. Transfection of siRNA Probes

HeLa cells seeded on 12 or 6 well plates were transfected with commercial siRNA probes (Ambion, Sydney, Australia) using Lipofectamine RNAiMax (Invitrogen, Waltham, MA, USA) following manufacturer’s recommendations. All siRNA probes are Silencer Select^TM^ (Ambion): Negative Control #1, 4390844; NFYA, s9530; SP1, s13319.

### 4.5. Purification of RNA, Retrotranscription and qPCR

Total RNA was isolated from approximately 10^6^ HeLa cells using the miRNeasy Mini Kit (Qiagen) following the manufacturer’s instructions. Total RNA quantity and quality (260/280 absorbance ratio) were assessed using NanoDrop 2000 (Thermo Fisher Scientific, Waltham, MA, USA).

One microgram of RNA was retrotranscribed using a High-Capacity RNA-to-cDNA Kit (Applied Biosystems, Foster City, CA, USA) following manufacturer’s recommendations and using a Veriti thermal cycler device (Applied Biosystems).

To assess the relative level of each studied mRNA, qPCR was performed using TaqMan™ Gene Expression Assays indicated below (Thermo Fisher Scientific), with TaqMan Gene Expression Master Mix (Applied Biosystems) and a LightCycler 480 device (Roche, Ludwigsburg, Germany) with the standard temperature cycles (annealing + extension = 1 min/60 °C). Normalization was performed with HPRT1. Relative expression was calculated as 2^−ΔΔCt^. All TaqMan probes (Applied Biosystems) have been validated. The probes are: HPRT1, Hs02800695_m1; NFYA, Hs00957373; SP1, Hs00916521_m1; PNPT1, Hs01105971_m1.

### 4.6. Transfection of Reporter Constructs and Luciferase Reporter Assay

Luciferase reporter assays were performed with HeLa cells. Transfections were carried out with FuGENE (Promega), following the manufacturer’s protocol. The day before transfection, HeLa cells were plated onto 96-well tissue culture dishes at a density that allowed cells to reach 60–70% confluence by the time of transfection (6000–7000 cells per well). Fresh culture media without antibiotics was used. The maximal activity of each reporter construct determined transfecting growing amounts (dose-response: 10—200 ng per well).

Luciferase activities were determined using a Victor X5 luminometer (Perkin Elmer), 0.5-s measuring period. The activities were expressed as relative values of the maximal activity, considering the activity of the wild-type promoter construct (396bp-PNPT1-prom-luc) as 100% and the activity of an empty vector (without promoter) as 0%.

Co-transfections of 396bp-PNPT1-prom-luc and siRNA probes were performed in 96-well plates using Lipofectamine 2000 (Invitrogen). Luciferase activity was assayed as described above. Silencing efficacies were assessed in parallel assays using 12-well plates.

### 4.7. Electromobility Shift Assays

Single-strand oligonucleotides were synthesized (IDT, Leuven, Belgium) for each region of the promoter and annealed to generate the probes. Probes were labeled with biotin for detection. Unlabeled probes were used in competition assays (Table 2). Preparation of the probes was carried out according to manufacturer’s instructions. Briefly, just before the experiment, complementary single-strand oligonucleotides (with or without biotin) were dissolved in DNAase-free water to a final concentration of 20 M in annealing buffer (60 mM KCl, 6 mM HEPES pH 7.5 and 0.2 mM MgCl_2_) in a total volume of 100 μL, incubated for 1 min at 95 °C and gradually cooled to room temperature. EMSA experiments were carried out using wild-type and mutated probes for the different putative transcription factor binding sites (Table 2). EMSAs for *PNPT1* promoter probes were performed using HeLa nuclear extract (HeLaScribe^®^ Nuclear Extract, Promega). EMSAs were carried out as follows: 6 μg of commercial HeLa nuclear extract were preincubated for 10 min at 4 °C in binding buffer, then incubated with an excess of non-labeled wild-type or mutated probes for 10 min at 4 °C, and finally during 30 min with 10 fmol of annealed biotin-labeled probes. The binding buffer was composed of 10 mM Tris-HCl pH 7.0, 50 mM KCl, 1 g of poly dI·dC, 5% glycerol, 1 mM EDTA and 1 mM DTT. For supershifts, 2 μg of antibody targeting NFYA (Santacruz, Heildeberg, Germany), SP1 (Santacruz), NRSF (Santacruz), or IRF-2 (Santacruz) were added and incubated for 30 additional min at 4 °C. Afterward, samples were loaded onto a 4% polyacrylamide (40:1, acrylamide:bisacrylamide), 0.5× tris-borate-EDTA (TBE) buffer gel and electrophoresed at 320 V, 4 °C in 0.5× TBE buffer. Electrophoretic blotting to a nitrocellulose membrane (Hybond™-N+, GE Healthcare) was carried out in semi-dry conditions (30 V) for 15 min in 0.5× TBE, followed by crosslinking under UV light (1250 J, 1 min). Membranes were blocked for 15 min in blocking buffer [5% sodium dodecyl sulfate (SDS) in phosphate saline buffer, pH 7.0] followed by incubation with horseradish peroxidase (HRP)-labeled streptavidin at 125 pg/mL for 15 min in the same buffer. Blots were washed 5 times with a 10-fold dilution of blocking buffer. Signals on the membranes were visualized by chemiluminescence detection (Las4000, GE Health Care, Grens, Switzerland).

### 4.8. Chromatin Immunoprecipitation

Chromatin immunoprecipitations (ChIPs) were performed in HeLa cells using the Pierce™ Magnetic ChIP Kit (ThermoScientific). Briefly, cells were crosslinked with 1% formaldehyde (Thermo Scientific) for 10 min and the kit manufacturer’s instructions followed. Chromatin was sheared enzymatically followed by sonication (30 s at 15% amplitude on ice, using an S-250 digital sonifier cell disruptor, Branson Ultrasonics, Danbury, CT, USA), as instructed by the kit. Immunoprecipitations were performed using 2 μg of the followiong antibodies: mouse monoclonal anti-NFYA antibody (Santa Cruz Biotechnology, sc-17753), rabbit policlonal anti-SP1 antibodies (Santa Cruz Biotechnology, sc-14027), normal mouse IgG antibody (EMD Millipore, *12*–*371*) or rabbit policlonal IgG antibodies provided in the kit. Real-time PCR over the ChIP products was performed in a LightCycler 480^®^ thermocycler (Roche) using a commercial polymerase mix (PowerUp™ SYBR^®^ Green Master Mix, Thermo Scientific) in triplicate. Enrichment as a percentage of the input was calculated and those values were referenced to the unspecific enrichment observed in the intergenic region. The primers used were: ON-PNPT1-PROM-3F, 5′-CACCGCGGAAACGAAACTC-3′ and ON-PNPT1-PROM-3R, 5′-CAGGCTGTGCCCTGATTGG-3′, to amplify the *PNPT1* promoter; and ON-CN-1F 5′-CTCTGTGCTGATACCTGGAGTCT-3′ and ON-CN-1R 5′-GGAAATGGGGGCCTATGTTTTGG-3′ to amplify the intergenic region.

### 4.9. Kaplan–Meier Survival Analysis

Data were downloaded from the Human Protein Atlas (data available from v21.proteinatlas.org searching the indicated genes; all links are shown in Appendix B).

Cancer samples of The Cancer Genome Atlas (TCGA) project had been analyzed by RNA-Sequencing (RNA-Seq). The mRNA expression levels were determined as FPKMs (number Fragments Per Kilobase of exon per Million reads). Based on the FPKM value of each gene, patients were classified into two expression groups and the correlation between expression level and patient survival was examined. The prognosis of each group of patients was examined by Kaplan–Meier survival estimators, and the survival outcomes of the two groups were compared by log-rank tests.

### 4.10. Statistical Analysis

Prism 5.0 software (GraphPad, San Diego, CA, USA) was used for all calculations. Gaussian distribution was determined with Shapiro–Wilk or Kolmogorov–Smirnov tests. Bars (and whiskers) indicate mean + SD. The statistical significance of two groups was determined using the unpaired *t*-test, and more than two groups were analyzed using one-way ANOVA followed by the post hoc Tukey test. Differences between two conditions were considered statistically significant at *p* < 0.05 (* *p* < 0.05; ** *p* < 0.01; *** *p* < 0.001).

## 5. Conclusions

In summary, we have characterized the promoter of the human *PNPT1* gene and established the essential roles of SP1 and NFY transcription factors in *PNPT1* expression. Our results suggest that SP1 and NFY binding sites work as a single cis-element within the *PNPT1* promoter, and that in hepatocellular cancer overexpressing PNPase, the promoter activity of *PNPT1* depends on SP1 and/or NFY.

## Figures and Tables

**Figure 1 ijms-23-11399-f001:**
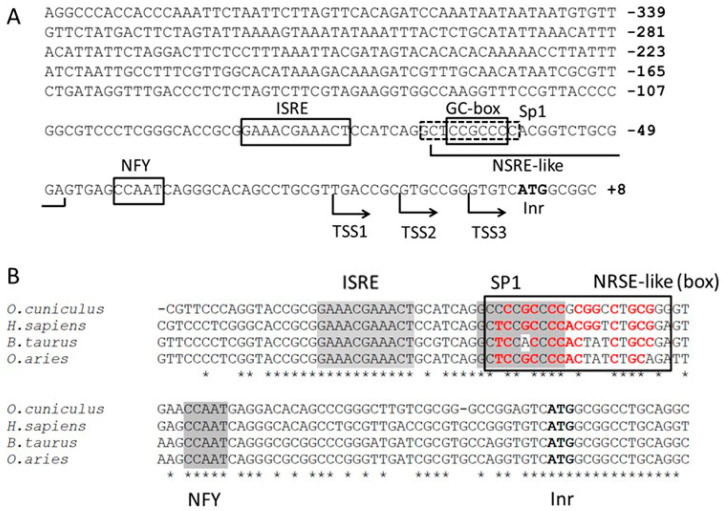
Putative motifs of the human *PNPT1* promoter and its conservation across species. (**A**) Nucleotides 4692 to 5095 of NG_033012.1 NCBI Reference sequence were analyzed with MatInspector (Genomatix) and Promo (Alggen) online applications. ISRE, GC-box and NFY binding site are highlighted with solid boxes, and putative SP1 binding site with a dotted box. NSRE-like sequence is underlined, and initial ATG (Inr) is shown in bold. Previously reported Transcription Start Sites (TSS) are shown. (**B**) Nucleotides 4984 to 5103 of NG_033012.1 NCBI Reference sequence corresponding to the human (*H. sapiens*) *PNPT1* promoter were aligned with homologous sequences of rabbit (*O. cuniculus*), cow (*B. taurus*) and sheep (*O. aries*), using Clustal Omega software (EMBL-EBI) with default settings. Animal sequences were previously retrieved with the Ensembl Genome Browser by means of a BLAST search using the human sequence as query. Nucleotides matching INRS, SP1 and NFY binding consensus sequences are shown with grey background. NRSE-like sequences are shown boxed, and nucleotides matching the reported NSRE consensus sequences are highlighted with red bold characters. Asterisks indicate nucleotides conserved in the four sequences.

**Figure 2 ijms-23-11399-f002:**
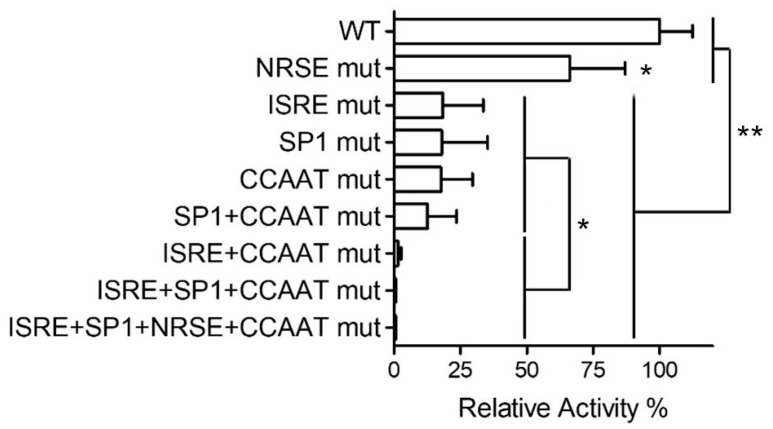
Mutational analysis of the predicted motifs of the human *PNPT1* promoter. HeLa cells were transiently transfected with a luciferase reporter plasmid containing 396 bp (nucleotides 4692–5087 of NG_033012.1 NCBI Reference sequence) of the *PNPT1* promoter (396bp-PNPT1-prom-luc) 5′ upstream from the ATG (WT, wild-type), or derived plasmids containing deleterious point mutations of the indicated motifs. The activity of the reporter was assayed 40 h after transfection. We represent the relative activity with respect to the native version of the promoter (WT). Statistical analysis: 1 way ANOVA /Tuckey: * *p* < 0.05; ** *p* < 0.01; *n* = 6 (3 assays on duplicated).

**Figure 3 ijms-23-11399-f003:**
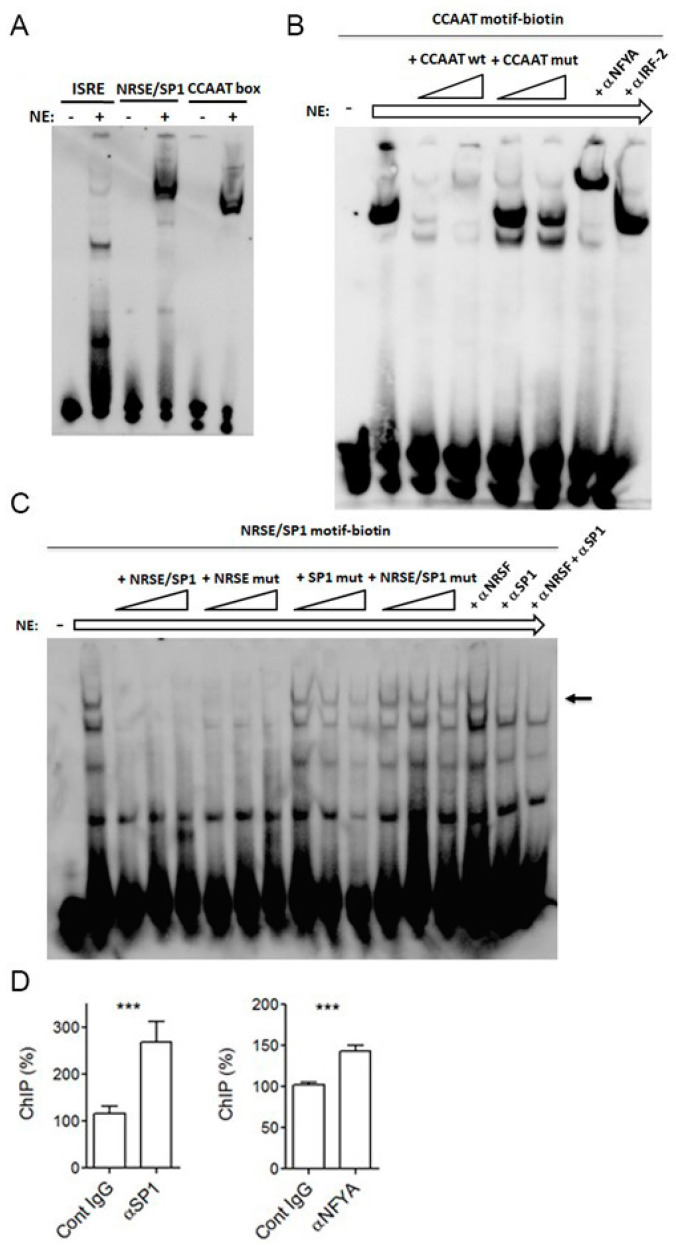
Mapping transcription complex binding sites in the 400-bp *PNPT1* promoter using EMSA and ChIP. (**A**) biotinylated double-stranded oligonucleotides representing the indicated cis-elements of 400 bp-*PNPT1* promoter were used for EMSA assays. SP1 and NRSE-like motifs overlap and were present in a single oligonucleotide. DNA fragments were incubated in the presence (+) or in the absence (–) of nuclear extracts (NE), and the resulting complexes were analyzed by nondenaturing PAGE and biotin detection. (**B**,**C**) the complexes observed in (**A**) were further competed with, either, growing concentrations of the identical unlabeled oligonucleotide or the corresponding point mutated counterpart, or antibodies to the indicated polypeptides. (**C**) the arrow points out the complex that is competed with the unlabeled oligonucleotide containing a native SP1 motif or with αSP1 antibodies but not with the native NRSE motif or the presence of αNRSE antibodies. Shown are representative EMSA. (**D**) SP1 or NFYA chromatin immunoprecipitation in HeLa cells. Immunoprecipitation was performed with antibodies to SP1 or NFYA and the corresponding unspecific control antibody (Cont IgG). Enrichment relative to controls was determined by real-time PCR and expressed as a percentage (fold) over control. Shown are representative ChIP assays performed in triplicate (mean ± SD). Two different pairs of oligonucleotides encompassing the promoter region yield similar results. Statistical analysis: *t*-test, *** *p* < 0.001 (*n* = 6 independent IP).

**Figure 4 ijms-23-11399-f004:**
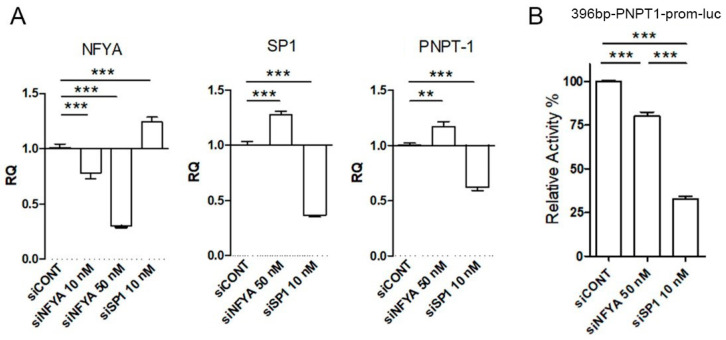
SP1 and NFYA expression effect on *PNPT1* mRNA levels and promoter activity. (**A**) relative mRNA levels of NFYA, SP1 and *PNPT1* were assessed by qPCR in HeLa cells treated with the indicated concentrations of a silencer to SP1 or NFYA (siSP1, siNFYA) or a commercial control silencer (siCONT). (**B**) HeLa cells were co-transfected with the 396 bp *PNPT1* prom-luc plasmid and with the indicated silencer. The luciferase activity was measured in cell lysates and expressed as relative units respect to control cells (mean ± SD). Statistical analysis: 1 way ANOVA/Tuckey: ** *p* < 0.01, ***, *p* < 0.001; (**A**) *n* = 9 (3 assays on triplicated); (**B**) *n* = 8 (2 assays on quadruplicated).

**Figure 5 ijms-23-11399-f005:**
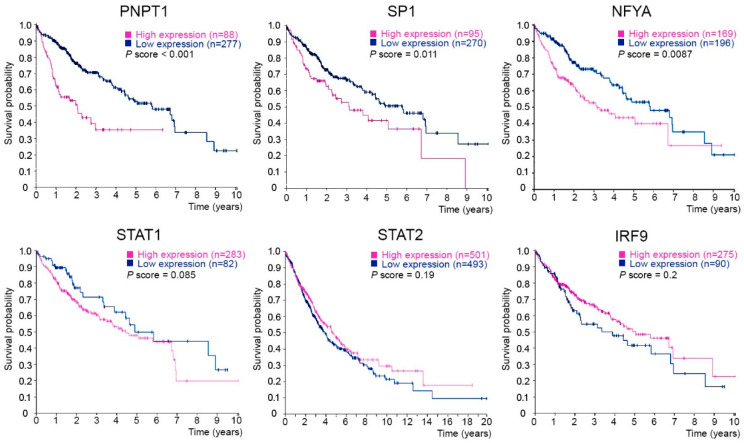
Increased expression of *PNPT1*, *SP1* and *NFYA* is associated with a poor prognosis in liver cancer. Shown are the Kaplan–Meier survival curves of liver cancer patients, classified as High or Low expression individuals depending on the expression levels of the indicated mRNA species in analyzed cancer tissues. Data were downloaded from Human Protein Atlas (data available from v21.proteinatlas.org searching the indicated genes; all links are shown in Appendix B).

**Table 1 ijms-23-11399-t001:** Wild-type and mutant sequences within nucleotides −96 to −1 of the *PNPT1* promoter region used in this study. Mutated nucleotides are shown in red.

Construct	Nucleotides −96 to −1 (Mutations in Red)
WT	GGGCACCGCGGAAACGAAACTCCATCAGGCTCCGCCCCACGGTCTGCG GAGTGAGCCAATCAGGGCACAGCCTGCGTTGACCGCGTGCCGGGTGTC
ISRE mut	GGGCACCGCGCAACTAGAACTCCATCAGGCTCCGCCCCACGGTCTGCG GAGTGAGCCAATCAGGGCACAGCCTGCGTTGACCGCGTGCCGGGTGTC
SP1 mut	GGGCACCGCGGAAACGAAACTCCATCAGGCTCCGAACCACGGTCTGCG GAGTGAGCCAATCAGGGCACAGCCTGCGTTGACCGCGTGCCGGGTGTC
NRSE mut	GGGCACCGCGGAAACGAAACTCCATCAGGCTCCGCCCCAAAATCTGCG GAGTGAGCCAATCAGGGCACAGCCTGCGTTGACCGCGTGCCGGGTGTC
CCAAT mut	GGGCACCGCGGAAACGAAACTCCATCAGGCTCCGCCCCACGGTCTGCG GAGTGAGCCAAACAGGGCACAGCCTGCGTTGACCGCGTGCCGGGTGTC
CCAAT+SP1 mut	GGGCACCGCGGAAACGAAACTCCATCAGGCTCCGAACCACGGTCTGCG GAGTGAGCCAAACAGGGCACAGCCTGCGTTGACCGCGTGCCGGGTGTC
ISRE+NFYA mut	GGGCACCGCGCAACTAGAACTCCATCAGGCTCCGCCCCACGGTCTGCG GAGTGAGCCAAACAGGGCACAGCCTGCGTTGACCGCGTGCCGGGTGTC
ISRE+SP1+NFYA mut	GGGCACCGCGCAACTAGAACTCCATCAGGCTCCGAACCACGGTCTGCG GAGTGAGCCAAACAGGGCACAGCCTGCGTTGACCGCGTGCCGGGTGT
ISRE+SP1+NRSE+NFYA mut	GGGCACCGCGCAACTAGAACTCCATCAGGCTCCGAACCAAAATCTGCG GAGTGAGCCAAACAGGGCACAGCCTGCGTTGACCGCGTGCCGGGTGTC

**Table 2 ijms-23-11399-t002:** Oligonucleotides used in EMSA. Motifs are shown in green and mutations in red.

Site(s)	Name	Sequence (5′-3′)
ISRE	PNPT1-ISRE-F PNPT1-ISRE-R	CACCGCGGAAACGAAACTCCATCAGG CCTGATGGAGTTTCGTTTCCGCGGTG
	PNPT1-ISRE-mut-F PNPT1-ISRE-mut-R	CACCGCGCAACTAGAACTCCATCAGG CCTGATGGAGTTCTAGTTGCGCGGTG
SP1/NRSE	PNPT1-SP1/NRSE-F PNPT1-SP1/NRSE-R	CATCAGGCTCCGCCCCACGGTCTGCGGA TCCGCAGACCGTGGGGCGGAGCCTGATG
	PNPT1-SP1-mut-F PNPT1-SP1-mut-R	CATCAGGCTCCGAACCACGGTCTGCGGA TCCGCAGACCGTGGTTCGGAGCCTGATG
	PNPT1-NRSE-mut-F PNPT1-NRSE-mut-R	CATCAGGCTCCGCCCCAAAATCTGCGGA TCCGCAGATTTTGGGGCGGAGCCTGATG
	PNPT1-NRSE+SP1-mut-F PNPT1-NRSE+SP1-mut-R	CATCAGGCTCCGAACCAAAATCTGCGGA TCCGCAGATTTTGGTTCGGAGCCTGATG
CCAAT box	PNPT1-CAATbox-F PNPT1-CAATbox-R	GGAGTGAGCCAATCAGGGCACAGCCTGC GCAGGCTGTGCCCTGATTGGCTCACTCC
	PNPT1-CAATbox-mut-F PNPT1-CAATbox-mut-R	GGAGTGAGCCAAACAGGGCACAGCCTGC GCAGGCTGTGCCCTGTTTGGCTCACTCC

## Data Availability

Not applicable.

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
