# Peer review of "SP1 and NFY Regulate the Expression of PNPT1, a Gene Encoding a Mitochondrial Protein Involved in Cancer†"

_ijms, 2022, doi:10.3390/ijms231911399_

Round 1
Reviewer 1 Report
The possibility that SP1 and NFY are involved in the regulation of PNPase was already known from promoter analysis of PNPase, therefore Figure 1-3 are just validation experiments to verify this possibility.
Figure 4 shows the experiment of knock down of SP1 and NFY by siRNA, but only one type of siRNA is used, so multiple siRNA experiments are needed to avoid off target effects. Especially for NFY, there is a big discrepancy between the PNPase qPCR results and the luciferase assay results, and they should not discuss the function of NFY without experiments using different siRNAs. In terms of robustness of the results, they are very weak.
Figure 5 shows an experiment on mitochondrial function, but mitochondrial function should not be discussed based only on a reduction study of resazurin, Respirometry (OXPHOS and Glycolysis), mitochondrial ROS, mitochondrial membrane potential, mitochondrial DNA copy number, mitochondrial biogenesis, and mitophagy, etc. should be measured and discussed. Also, we are most interested in whether the reduction in PNPase affected RNA import into mitochondria, but no experiments have been done on this point.
Figure 6 is an interesting result from the Database, but it is only an indirect evidence of PNPase involvement; if there is a change in allover survival or metastasis in transplantation experiments of tumor cells with low PNPase expression, it would be direct evidence. I believe that the authors' argument is overestimate.
Reviewer 2 Report
The authors that examined the PNPT1 gene, which encodes for polynucleotide phosphorylase (PNPase) and those involved in RNA processing and degradation, mainly located in the inter-membrane space and involved in RNA import into the mitochondrion. They are defined PNPT1gene promoter activity and its possible binding transcriptional factor SP1 or NFY. Final, they have checked gene function in cancer cells. Some comments is as following:
1. In Introduction should mention PNPT1 gene its location in which one chromosome.
2. In figure 6, has showed the liver cancer in patients, but did not mention in Material and Methods.
3. The authors should provide a summary figure to summarize the results for more ease to check the point.
4. In the figure 4 and 5 for SP1 or NFY functional assay, by using loss-of-function or gain-of-function assays to examine and got the SP1 or NFY can directly to regulate the PNPT1 gene in cancer cells. As I known SP1 or NFY did not just regulate one single gene, so sure have more tightly design in this part of function.
Reviewer 3 Report
The authors of the publication conducted a functional analysis of the PNPT1 gene promoter, focusing on 3 previously identified transcription factor binding sites, two of which have not been confirmed experimentally. The authors manage to demonstrate, using a luciferase reporter, that the PNPT1 promoter is regulated by two independent transcription complexes - ISRE and SP /NFY.
The authors should respond to the following comments.
The role of PNPase in the import of RNA into the mitochondria does not appear to be the primary function of this protein. This information should be removed from the title of the publication, the more so that the presented results are not related to this issue.
It has been clearly shown that human mitochondrial RNAseP is a protein-only complex with no RNA component. The information provided on this subject in the introduction misleads the reader.
The authors do not cite any publication on the involvement of PNPase in the degradation of RNA in human mitochondria, such as PMID: 22661577 and PMID: 23221631, but cite a publication that deals with the function of PNPase in the degradation of oxidized RNA in bacteria as if it were human protein.
The role of PNPase in the degradation and surveillance of mitochondrial RNA seems to be the main and well-documented function of PNPase in various eukaryotic organisms, therefore I suggest moderation in emphasizing the role of this protein in RNA import into mitochondria, because for many scientists it is a quite controversial topic.
The end of subsection 2.4, starting with the sentence: The apparent discrepancy between the effect of NFYA, should be moved to the Discussion section.
I do not see the sense of the experiment described in section 2.5 and shown in Figure 5. The authors in the discussion cite literature that shows that SP1 and NFY are involved in the expression of mitochondrial proteins, so it is not unusual that silencing these genes affects function mitochondria. If the authors want to associate this result with a decrease in PNPase level, I expect the results of a rescue experiment.
The first paragraph of the Discussion section is completely irrelevant to the presented results, it should be removed.
The Materials and Methods lack information about the siRNAs used (catalog numbers, sequences), and there is also a lack of information about the probes used for qPCR.
Round 2
Reviewer 1 Report
I dose not consider your revise response my concerns.
Author Response
Point 1: The possibility that SP1 and NFY are involved in the regulation of PNPase was already known from
promoter analysis of PNPase, therefore Figure 1-3 are just validation experiments to verify this possibility.
Response 1: The possibility that SP1 was involved in PNPase regulation was just a prediction. We have not
found published reports of a similar predicted function proposed for NFY.
Predictions do not mean that the activity is relevant for a certain cell type and growing condition, as we show
for NFY: in our case, the levels of NFY are not critical for PNPT1 promoter activity whereas SP1 is necessary for
its activity.
As the reviewer states we demonstrate the predicted activities. These activities had not been previously reported
and we introduce interesting nuances. These results allow to establish conclusions and suggest additional ideas
that could help to readers to understand how this promoter works:
CONCLUSIONS
1) SP1 binds the PNPT1 promoter in the predicted cis SP1 motif in HeLa cells.
2) The binding of SP1 activates the promoter in HeLa cells cultured with standard media and conditions.
3) NFYA (NFY) binds the PNPT1 promoter in a cis NFY motif. The presence of this motif was not
proposed in the latest reviews and research articles.
4) NFYA levels are not critical for the transcriptional activity of the PNPT1 promoter of HeLa cells
cultured with standard media and conditions.
5) NFY site is relevant for PNPT1 promoter activity.
SUGGESTIONS
6) The point mutation studies suggest that SP1 and NFY sites work as a single cis element. The second
independent transcriptional unitcis element is the ISRE site which had been previously described.
7) Kaplan-Meier survival curves suggest that in hepatocellular cancer in vivo ‘growing conditions’ , the
promoter activity of PNPT1 in patients overexpressing PNPase depends on SP1 and/or NFY. The
characterization of this microenvironment or ‘growing conditions’ could be a starting point for further
research.
We have introduced a CONCLUSIONS section to summarize them.
Point 2: Figure 4 shows the experiment of knock down of SP1 and NFY by siRNA, but only one type of siRNA
is used, so multiple siRNA experiments are needed to avoid off target effects. Especially for NFY, there is a big
discrepancy between the PNPase qPCR results and the luciferase assay results, and they should not discuss the
function of NFY without experiments using different siRNAs. In terms of robustness of the results, they are very
weak.
Response 2: We employed Silencer® Select validated siRNAs. They have been designed and incorporate
chemical modifications to reduce off-target activity.
2
In the case of SP1 siRNA, it was employed at 10 nM, a concentration in which a relevant off-target activity is
very unlikely1. Additionally, we have employed a second siRNA for SP1 that was previously used for the
characterization of the promoter of COL4A3BP2. The efficacy of the second siRNA was lower, and we did not
show it, however the results were in the same direction. Additionally, these results are in the line of the
characterization of the cis element shown here, and the transcriptional activity of SP1 shown in other articles.
In the case of NFYA the reduction of its levels did not produce a reduction of the mRNA of PNPT1, and
consequently, an off-target activity would not change our conclusions.
There is a certain discrepancy but we should consider several aspects that could explain them:
1) The silencing of NFYA does completely abolish its expression. The levels are 3 – 4 times lower but
enough to maintain the promoter activity.
2) The transfection of a reporter plasmid raises the amount of promoter copies per cell up to several
thousand times3.
3) Considering the different ratios, a a 20%-reduction of the promoter activity secondary to a 70%-
reduction of NFYA levels is not incompatible with absence of effect on PNPT1 mRNA levels secondary
to an equivalent silencing.
We have included a statement in the article.
Point 3: Figure 5 shows an experiment on mitochondrial function, but mitochondrial function should not be
discussed based only on a reduction study of resazurin, Respirometry (OXPHOS and Glycolysis), mitochondrial
ROS, mitochondrial membrane potential, mitochondrial DNA copy number, mitochondrial biogenesis, and
mitophagy, etc. should be measured and discussed. Also, we are most interested in whether the reduction in
PNPase affected RNA import into mitochondria, but no experiments have been done on this point.
Response 3: The main goal of this article is to characterize the cis and trans elements of PNPT1 promoter. Figure
5 is not critical for it. However, we wanted to show that NFY is relevant for mitochondrial activity despite it
does not affect PNPT1 levels. We think this result could help others. The mechanism is out of the scope of this
article.
Point 4: Figure 6 is an interesting result from the Database, but it is only an indirect evidence of PNPase
involvement; if there is a change in allover survival or metastasis in transplantation experiments of tumor cells
with low PNPase expression, it would be direct evidence. I believe that the authors' argument is overestimate.
Response 4: We agree that figure 6 does not provide a direct evidence of the role of PNPase in survival. We
pretended to suggest a role of NFY and SP1 in PNPase overexpression in a number of liver cancer patients, in
which this overexpression is a negative prognostic factor. We have included a statement in CONCLUSIONS.

Reviewer 2 Report
In this revision has improved much can consider for publish in the journal.
Author Response
solved.
Reviewer 3 Report
The authors responded satisfactorily to all comments.